# Research on the Thickness and Microstructure of Plate-like TiO$_2$ by the Nanosheet-Seeding Growth Technique

Yanyan Zhang [1,†], Hao Liu [1,†], Junyan Cui [1], Xiaosong Bai [1], Daoyuan Yang [1], Huiyu Yuan [1,2,*] and Baoming Wang [3,*]

1   Henan Key Laboratory of High Temperature Functional Ceramics, School of Materials Science and Engineering, Zhengzhou University, Zhengzhou 450001, China
2   Henan Institute of Product Quality Supervision and Inspection, Zhengzhou 450014, China
3   School of Ecology and Environment, Zhengzhou University, Zhengzhou 450001, China
*   Correspondence: hyyuan@zzu.edu.cn (H.Y.); baomingwang@zzu.edu.cn (B.W.)
†   These authors contributed equally to this work.

**Abstract:** The nanosheet-seeding growth (NSG) technique is an interesting synthesis method for preparing two-dimensional (2D) materials by employing ultrathin nanosheets as templates. In this work, the synthesis of 2D TiO$_2$ nanoplates using Ti$_{0.87}$O$_2$ nanosheets via the NSG process is thoroughly studied to achieve a better understanding of this process. The influence of various synthesis conditions on the morphology and phase composition has been carefully examined. The study of synthesis time reveals that the TiO$_2$ grows in the Stranski–Krastanov mode on the templates and the growth follows second-order kinetics. It is also found that the concentration of precursors and the synthesis time are the effective parameters in controlling the thickness of TiO$_2$ nanoplates. The phase of the sample changes from anatase TiO$_2$ to NH$_4$TiOF$_3$ and the morphology changes from flake to disk with the increase in the precursor concentration. The synthesis temperature has a large influence on the morphology and thickness of the sample but has little effect on the phase composition. However, the synthesis temperature changes the color of the sample, and a high temperature enlarges the light absorption range of the sample.

**Keywords:** nanosheet-seeding growth; template synthesis; TiO$_2$ nanoplate; NH$_4$TiOF$_3$; thickness control





## 1. Introduction

Two-dimensional (2D) materials have received tremendous attention due to their interesting physical, chemical, electronic and optical properties [1–4]. At present, all of the synthesis methods can be categorized into two categories: top-down and bottom-up methods. In the top-down strategy, nanosheets are exfoliated from their bulk layered crystals, either by chemical reaction or mechanical effect [5–9]. It is impossible to synthesize the 2D compounds without layered precursors by this method. The bottom-up strategy involves directly growing 2D materials starting from atoms, ions or molecules. It has no limitation on the prerequisite structure in contrast to the top-down method, but it is a great challenge to find a general synthetic route to obtain 2D materials with different compositions. Therefore, it is necessary to develop a new strategy to synthesize 2D materials.

The nanosheet-seeding growth (NSG) technique is a template growth method that combines the bottom-up and top-down strategies for the synthesis of 2D materials. Hua et al. synthesized graphene-like 2D-Al$_2$O$_3$ nanosheets by using graphene as seeds [10]. This method was used to synthesize 2D TiO$_2$ anatase crystals as well, and the TiO$_2$ anatase nanoplates can be tuned to expose the {001} or {100} facet by the selection of the proper nanosheet seeds, such as Ti$_{0.87}$O$_2$ and Ca$_2$Nb$_3$O$_{10}$. The TiO$_2$ anatase nanoplates seeding on Ti$_{0.87}$O$_2$ show high photocatalytic performance for water splitting [11,12]. However, controlling the morphology or microstructure of the nanoplates has not been studied, even though the morphology is one of the most important parameters that determine

the performance of the material [13]. On the other hand, the preparation process of the NSG method needs to be completed in a liquid phase under stirring conditions in order to achieve well dispersion of the nanosheet template. The turbulence could have a huge impact on the prepared samples, but little work has been carried out in this respect. The above-mentioned issues are greatly dependent on the crystal growth behavior of materials. Thus, it is important to investigate the crystal growth behavior of materials on ultrathin nanosheets.

$TiO_2$ is a promising photocatalytic material and has been widely studied. For example, artificial oxygen defects can be generated on its surface, which can improve the photocatalytic performance of $TiO_2$. It is found that Ag-$TiO_2$ nano-structured nanofibers have excellent structural and physical properties, which are suitable for efficient photocatalytic and antimicrobial applications [14,15]. Moreover, the influence of the poly(titanium oxide) structure has a significant effect on the relaxation behavior of organo-inorganic interpenetrating polymer network samples [16]. Ultrathin $TiO_2$ flakes can fully optimize their crucial $CO_2$ photo reduction processes by affording abundant catalytically active sites and increased two-dimensional conductivity, which could potentially help to relieve the increasing energy crisis and the worsening global climate [17]. In this paper, the synthesis of anatase $TiO_2$ nanoplates via the NSG technique has been studied thoroughly to understand the crystal growth of $TiO_2$ on ultrathin nanosheets. The crystal growth of $TiO_2$ was investigated by varying the synthesis conditions, including the precursor concentration, the synthesis temperature, the synthesis time and the stirring rates (turbulence), and the morphology and microstructure change were also discussed.

## 2. Experiment

### 2.1. Preparation of $Ti_{0.87}O_2$ Nanosheets

$Ti_{0.87}O_2$ nanosheets were prepared by the exfoliation of the layered titanate $K_{0.8}Li_{0.27}Ti_{1.73}O_4$ (KLTO) [18–20]. The first step is to prepare layered KLTO by high-temperature solid-phase sintering. The raw materials of $Li_2CO_3$ (Macklin, 99%, Shanghai, China), $K_2CO_3$ (Macklin, 99.5%) and $TiO_2$ (Macklin, 99%), with a molar ratio of 0.8: 0.27: 3.46, are mixed and calcined at 800 °C. After 30 min, the mixture was naturally cooled to room temperature, and the pre-fired mixture was taken out and ground in a mortar. Then, the mixture was calcined at 1200 °C for 24 h, and KLTO crystals were obtained after natural cooling. In the second step, KLTO crystals were further processed in HCl solution to prepare $H_{1.07}Ti_{1.73}O_4$ (HTO) crystals. Each gram of KLTO was treated with 100 mL of 1 mol/L HCl solution. The acidic solution was refreshed every day. After 3 days of the treatment, the powder was washed with a large amount of deionized water, filtered and naturally dried, and then the HTO powder was obtained. In the third step, 0.1 g of HTO powder was mixed with water and TBAOH, with a molar ratio of TBAOH/$H^+$ of 4/1, in a total volume of 20 mL. After stirring for 7 days, the $Ti_{0.87}O_2$ nanosheet suspension was obtained [21,22]. The nominal concentration of the $Ti_{0.87}O_2$ nanosheet suspension is 5 g/L.

### 2.2. Preparation of Anatase $TiO_2$ Nanoplates

Anatase $TiO_2$ nanoplates were synthesized by the nanosheet-seeding growth of $TiO_2$ on the $Ti_{0.87}O_2$ nanosheets, as described by the previous study [11,12]. Briefly, $(NH_4)_2TiF_6$ (Macklin, 98%) and $H_3BO_3$ (Macklin, 99.5%) were used as precursors, and they were dissolved in 200 mL deionized water while vigorously stirring. The molar ratio of $(NH_4)_2TiF_6/H_3BO_3$ was $\frac{1}{2}$, and the $(NH_4)_2TiF_6$ concentration was 0.05 M (or 0.025 M, 0.1 M, 0.15 M, 0.2 M and 0.3 M). After mixing the precursors for 1 min, 2.4 mL $Ti_{0.87}O_2$ nanosheets were used as templates to prepare the nanoplates. The mixture had been stirred at 30 °C (or 10, 50, 70, 80, 90 and 100 °C) for 48 h (or 3, 6,12 and 24 h) to allow the $TiO_2$ crystal to grow on the TO nanosheets. The stirring rate of the mixture was 900 rpm (or 300, 600, 1200 and 1500 rpm). The final products were filtered, washed and collected. The resulting powders were then annealed at 450 °C for 4 h in air.

### 2.3. Characterization

The X-ray diffraction (XRD) data of the samples were acquired on a Philips X'pert diffractometer with Cu$K\alpha$ radiation ($\lambda$ = 1.5418 Å). Scanning Electron Microscopy (SEM, ZEISS MERLIN compact, Oberkochen, Germany) was used to acquire information on the morphology, and the image software (version 2015) was used to measure the layer thickness. Each thickness datum was obtained from at least 10 different nanoplates. Transmission Electron Microscopy (TEM) was performed on a JEOL JEM-2100 at 200 kV (Tokyo, Japan), with the sample supported on a copper microgrid. UV-Vis diffuse reflectance spectra (UV-Vis DRS) were recorded with a Shimadzu (Tokyo, Japan) UV-3600 UV-Vis spectrophotometer in transmission mode within a 200–800 nm range.

## 3. Results and Discussion

By using the NSG technique, it is facile to obtain plate-like particles under various conditions, and the particles show similar morphological and element features. As shown in Figure 1a,b, the samples are a sheet structure and are covered with particulate matter. The lattice spacing of 0.24 nm and 0.35 nm in Figure 1c,d corresponds to the (103) and (101) planes of anatase type $TiO_2$, respectively, evidencing the formation of anatase $TiO_2$. The EDS analysis shows that all the samples consist of Ti, O and F elements, as shown in Figure 1e–h. Obviously, the contents of Ti and O distribute uniformly. In addition, the F content in all the samples under various conditions is ~1.1 mass%, which is due to the residual element from the precursor solution [23,24].

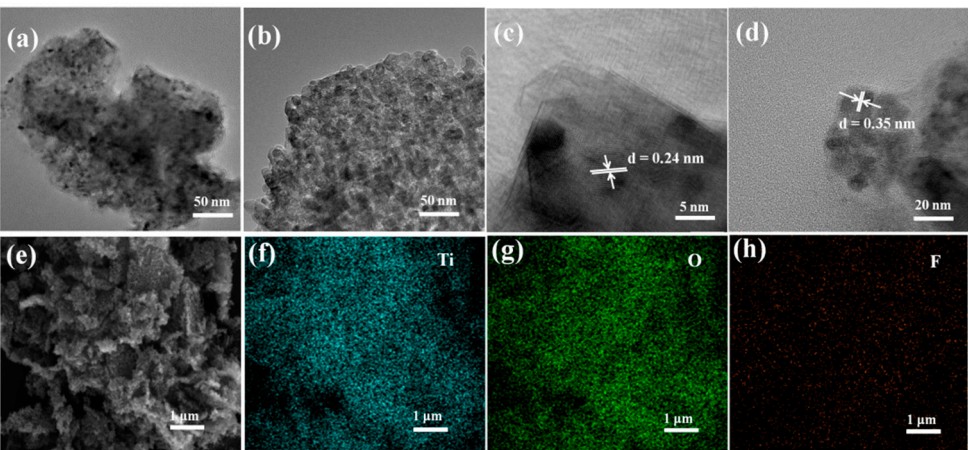

**Figure 1.** HRTEM images (**a–d**) and elemental mappings (**e–h**) of the plate-like $TiO_2$ samples.

### 3.1. The Effect of Synthesis Time

The effect of synthesis time on the formation of crystalline $Ti_{0.87}O_2$ nanoplates was studied at 30 °C with the $(NH_4)_2TiF_6$ concentration of 0.05 M. The XRD patterns of $Ti_{0.87}O_2$ nanoplates obtained at different synthesis times (Figure S1) illustrate that all samples are anatase $TiO_2$ (JCPDS 21-1272). In addition, all the samples have sharp peaks, indicating the good crystallinity of the samples. The SEM images of the $Ti_{0.87}O_2$ nanoplates at various synthesis times are shown in Figure 2. It can be clearly seen that the morphology of all the samples is sheet-like. However, the morphology changes with time. When the synthesis time was 3 h, the deposition was observed on the template as island-like (Figure 2a), and its thickness reached 20 nm, much thicker than the original nanosheets (Figure 2d). After 6 h, the surface of the prepared sample is almost completely covered with needle-like particles (Figure 2b), and the thickness was 55 nm (Figure 2e). After 12 h, the surface feature and thickness of the sample hardly change (Figure 2c,f–h). By fitting the thickness with the different kinetic modes, as presented in Table 1, it is found that the $TiO_2$ growth follows the second-order kinetics (with autocatalysis), as shown in Figure 2i. The fitting to this kinetics yields a reaction rate of $1.68 \times 10^{-6}$ s$^{-1}$.

　　　　The XRD patterns shown in Figure S1 show all the samples with a good crystallinity. It can be calculated by the Scherrer formula that the crystallite sizes with different synthesis times are 11.8 nm, 11.5 nm, 12.6 nm, 12.7 nm and 13.4 nm, respectively. The crystallite size increases slightly with the change in synthesis time. We employed the Johnson–Mehl–Avrami–Kolmogorov (JMAK) model to obtain the details of the crystal growth, and the fitting shown in Figure 3 yields an Avrami kinetic exponent n of 1.76. The Avrami kinetic exponent is a decimal between 1 and 2, suggesting that the crystallization process on nanosheets is a combination of one-dimensional and two-dimensional growth. In order to visualize the crystal growth of $TiO_2$ on nanosheets, the deposition was also conducted on Langmuir–Blodgett fabricated nanosheet thin films, and the surface configuration was studied by AFM. The AFM results (Figure S2) show that the thin film becomes thicker and thicker (Figure S2a–h), and the surface is smooth at the initial stage of the deposition (roughness: 0.17 nm at 2 h) and becomes rougher and rougher with time (Figure S2i–l). So, it is concluded that the growth of $TiO_2$ on the $Ti_{0.87}O_2$ nanosheet follows the Stranski–Krastanov growth mode [25–27].

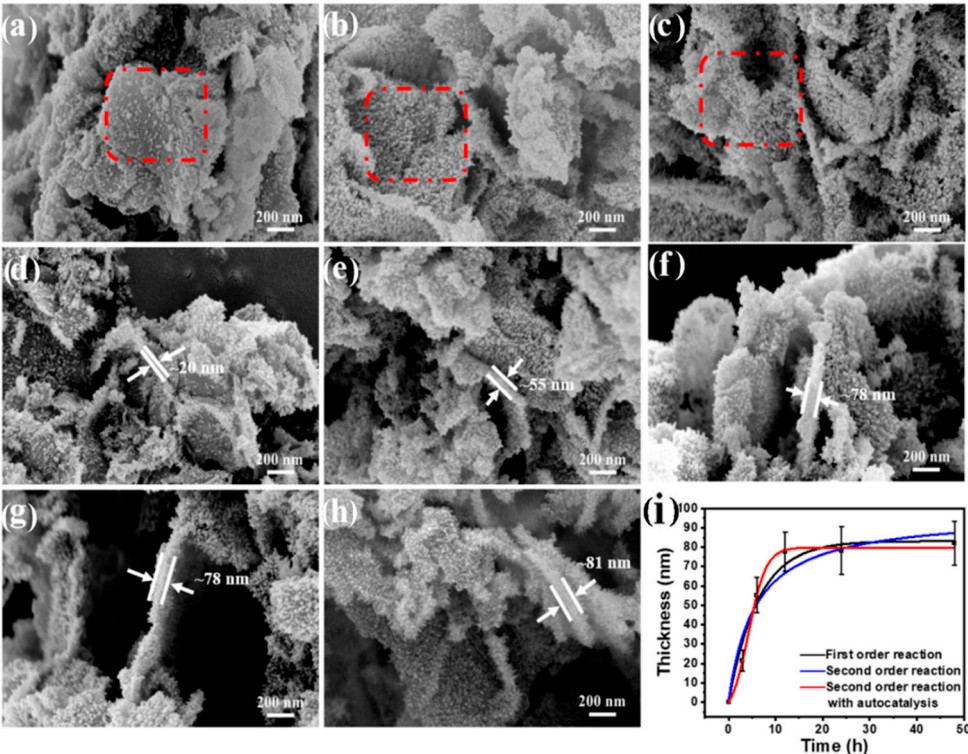

**Figure 2.** SEM images of the prepared samples with different synthesis times: (**a,d**) 3 h, (**b,e**) 6 h, (**c,f**) 12 h, (**g**) 24 h and (**h**) 48 h. (**i**) Fitting curves to the sample thickness. Red circles highlight the surface characteristics.

**Table 1.** Summary of the kinetic fitting to the sample synthesis.

| Reaction Kinetics | Kinetic Equation | *k* Value | $R^2$ |
|---|---|---|---|
| First-order reaction | $y = a\left(1 - e^{-kt}\right)$ | $k = 4.45 \times 10^{-5}\ s^{-1}$ | 0.97009 |
| Second-order reaction | $y = \frac{a^2kt}{1+akt}$ | $k = 5.16 \times 10^{-7}\ s^{-1}$ | 0.94305 |
| Second-order reaction with autocatalysis | $y = \frac{ab\left(e^{(a+b)kt}-1\right)}{a+be^{(a+b)kt}}$ | $k = 1.68 \times 10^{-6}\ s^{-1}$ | 0.99871 |

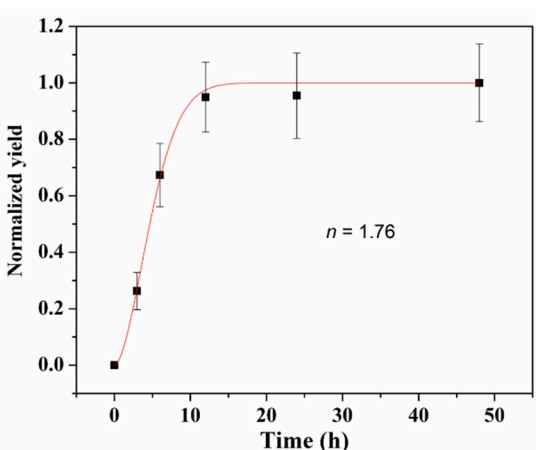

**Figure 3.** Fitting of the crystallization of $TiO_2$ on nanosheets by means of the Avrami equation (red line is the fitting curve).

### 3.2. The Influence of $(NH_4)_2TiF_6$ Concentration

The SEM images of the samples prepared with different $(NH_4)_2TiF_6$ concentrations from 0.025 M to 0.3 M are shown in Figure 4. When the concentration of $(NH_4)_2TiF_6$ is 0.025 M, the prepared sample stacks severely due to the small thickness (Figure S3a). When the concentration of $(NH_4)_2TiF_6$ is 0.05 M and up to 0.2 M, all the samples demonstrate a sheet-like structure, and the nanoplates are covered by vertically standing needle-like crystals (Figure S3b–e). The surface feature of the samples is consistent with our previous results [11]. It is noted that the disc-like structure was also observed in the sample with the $(NH_4)_2TiF_6$ concentration of 0.2 M. On the other hand, the thickness of the sample increases as the concentration of $(NH_4)_2TiF_6$ increases. A statistical study shows that the thickness of the nanoplates obtained is 65–130 nm (Figure 4f). In detail, the nanoplates are composed of a dense layer and a needle-like layer. The thicknesses of the dense layers of different samples are about 41 nm, 65 nm, 84 nm and 97 nm, respectively. In contrast to the dense layer, the thickness of the needle-like layer does not change significantly as the precursor concentration increases, and its range is 20–33 nm (Figure 4f). So, the overall thickness is mainly attributed to the increased dense layer. This feature reflects that the crystal growth of $TiO_2$ on nanosheets follows the Stranski–Krastanov growth mode, which is consistent with the previous discussion. When the concentration of $(NH_4)_2TiF_6$ increases further to 0.3 M, the morphology of the sample becomes a disc-like structure (Figure S3e). After annealing, the morphology of the sample changes from a disc-like structure to a regular cubic structure (Figure S3f).

Figure 5 shows the XRD patterns of the samples before and after calcination at 450 °C. When the concentration of $(NH_4)_2TiF_6$ is less than 0.2 M, the prepared sample is anatase $TiO_2$ (JCPDS 21-1272), as shown in Figure 5a. However, when the concentration of $(NH_4)_2TiF_6$ is 0.2 M, the phase of the prepared sample is $TiO_2$ and $NH_4TiOF_3$. As the concentration of $(NH_4)_2TiF_6$ further increases to 0.3 M, the phase of the prepared sample is only $NH_4TiOF_3$. This phenomenon is also reported in a previous report [28]. However, all the annealed samples are anatase $TiO_2$, and the prepared samples have good crystallinity (Figure 5b). When the concentration of $NH_4TiOF_3$ increases from 0.025 M to 0.3 M, after calcination, the crystallite sizes of $TiO_2$ are 15.5 nm, 15.0 nm, 12.7 nm, 13.4 nm, 15.1 nm and 46.4 nm, respectively. The much larger crystallite size of the latter one can be explained by the disc-like structure, while the others are plate-like structures (Figure 4e).

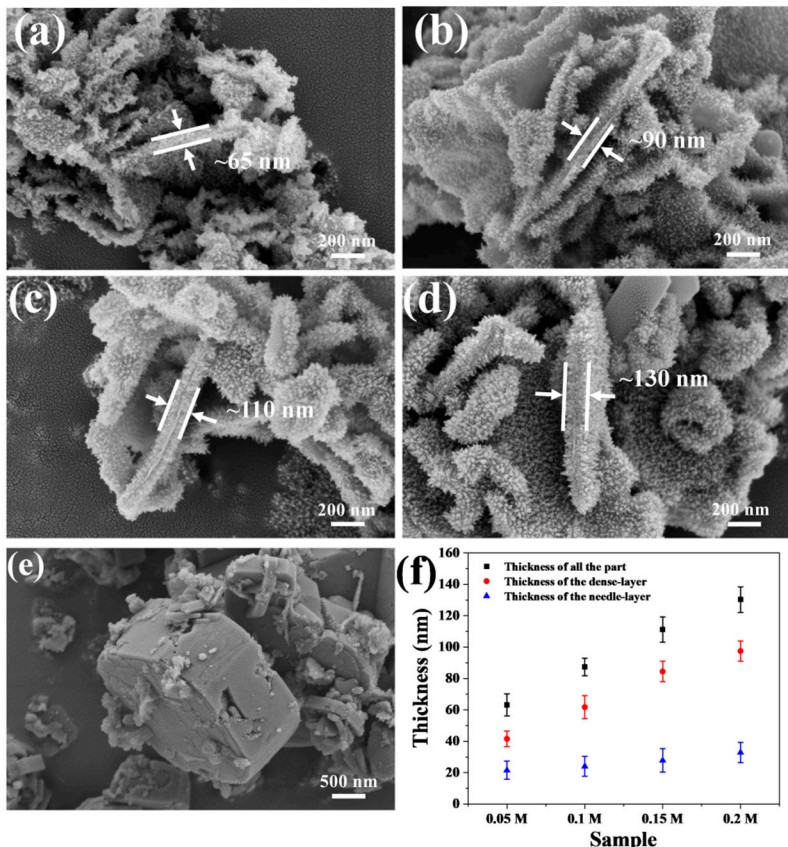

**Figure 4.** SEM images of the samples prepared with different $(NH_4)_2TiF_6$ concentrations before calcination: (**a**) 0.05 M, (**b**) 0.1 M, (**c**) 0.15 M, (**d**) 0.2 M, (**e**) 0.3 M. (**f**) The thickness data of the samples.

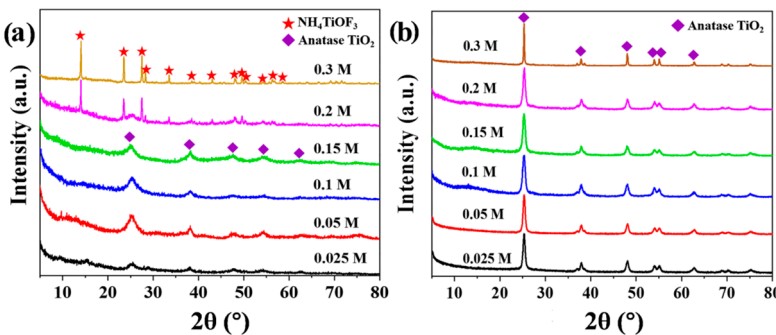

**Figure 5.** XRD of the different samples (**a**) before and (**b**) after annealing.

The $TiO_2$ deposition on the template can be described by the following processes: [29,30]

$$[TiF_6]^{2-} + nH_2O \rightleftharpoons [TiF_{6-n}(OH)_n]^{2-} + nHF \qquad (1)$$

$$H_3BO_3 + 4HF \rightleftharpoons BF_4^- + H_3O^+ + 2H_2O \qquad (2)$$

$$[Ti(OH)_6]^{2-} \rightarrow TiO_2\downarrow + 2H_2O + 2OH^- \qquad (3)$$

Apparently, the addition of $H_3BO_3$ causes the consumption of non-coordinating $F^-$ ions in Equation (2), accelerates the hydrolysis reaction of $[TiF_6]^{2-}$ to produce $[Ti(OH)_6]^{2-}$ in Equation (1) and finally dehydrates $[Ti(OH)_6]^{2-}$ to produce $TiO_2$ in Equation (3). In the process of preparing $TiO_2$ with $(NH_4)_2TiF_6$ and $H_3BO_3$, $NH_4TiOF_3$ is also produced. The reaction equations can be described as follows [31]:

$$[TiF_6]^{2-} + 3H_2O \rightleftharpoons [TiF_3(OH)_3]^{2-} + 3HF \qquad (4)$$

$$[TiF_3(OH)_3]^{2-} + H^+ + NH_4^+ \rightleftharpoons NH_4TiOF_3\downarrow +3H_2O \qquad (5)$$

The hydrolysis of $[TiF_6]^{2-}$ is a gradual process, as shown in Equations (1) and (2), and it cannot be hydrolyzed in one step to produce $[Ti(OH)_6]^{2-}$. In its hydrolysis process, $[TiF_3(OH)_3]^{2-}$ can be generated as an intermediate, as stated in Equation (4). When the concentration of $(NH_4)_2TiF_6$ is high compared to the concentration of $HBO_3$ in the solution, the hydrolysis of $[TiF_6]^{2-}$ can produce more $[TiF_3(OH)_3]^{2-}$, which consequently combines with $NH_4^+$ and $H^+$ to form the $NH_4TiOF_3$ in the solution, as shown in Equation (5) [32–34].

### 3.3. The Influence of Stirring Rate

The effect of the stirring rate on the material growth is also studied in this work. Different samples were prepared at stirring rates between 300 rpm and 1500 rpm, while keeping the rest of the process parameters the same. Figure 6 shows the SEM images of different samples, and the results reveal that different samples have a similar surface morphologies and that all the samples are sheet-like with needle-like crystals on the surface (Figure S4a–e). The statistical analysis of the thickness of the sample reveals that its thickness did not change with the change in the stirring rate, and the average thickness of different samples was about 80 nm, as shown in Figure 6f.

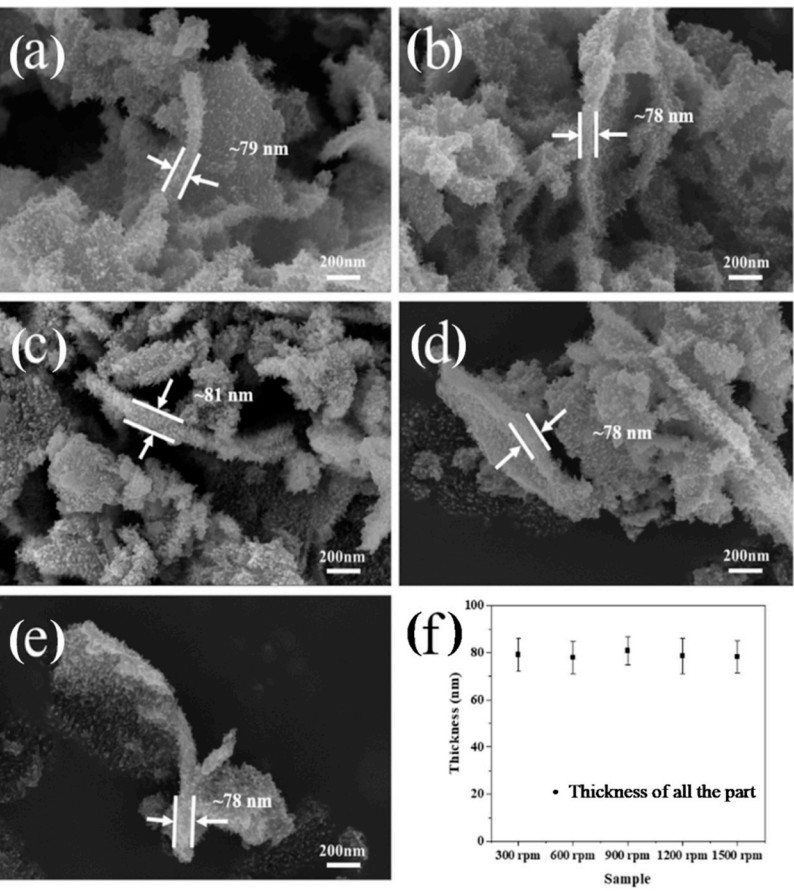

**Figure 6.** SEM images of the prepared samples with different stirring rates: (**a**) 300 rpm, (**b**) 600 rpm, (**c**) 900 rpm, (**d**) 1200 rpm and (**e**) 1500 rpm. (**f**) The thickness data of the samples.

The XRD patterns of the samples shown in Figure S4f demonstrate that all the samples have the same peak positions, and the phase composition is identified to be anatase $TiO_2$ (JCPDS 21-1272). The crystallite size is 13.8, 13.1, 14.0, 14.2 and 13.5 nm, respectively, and the stirring rate has little effect on the crystallite size. On the other hand, the peak intensities of all the samples are strong, suggesting a good crystallinity.

### 3.4. The Effect of the Synthesis Temperature

In order to investigate the effect of temperature on the morphology and thickness of the samples, the samples with synthesis temperatures from 10 °C to 100 °C were synthesized while other conditions were kept identical. The SEM images of the different samples are shown in Figure 7a–e, which demonstrate that all the samples have a sheet-like structure, and the surface is covered by spike-like crystals. As the synthesis temperature increases, the crystals on the surface of the sample become dense, and the morphology of the crystals changes from needle-like to hump-like. The statistics of the thickness of different samples, shown in Figure 7f, reveal that the thickness of the sample increases as the temperature increases, from ~55 nm at 10 °C to ~90 nm above 50 °C. Furthermore, the XRD patterns shown in Figure S5 evidence that all of the samples are anatase $TiO_2$. The crystallite size is 11.5 nm, 13.4 nm, 15.3 nm, 16.7 nm, 18.1 nm, 17.3 nm and 17.2 nm, respectively. Before 70 °C, the peak intensity of the XRD patterns enhances with the increase in temperature, and the crystallite size is also increasing. When the synthesis temperature exceeds 70 °C, the crystallite size changes slightly.

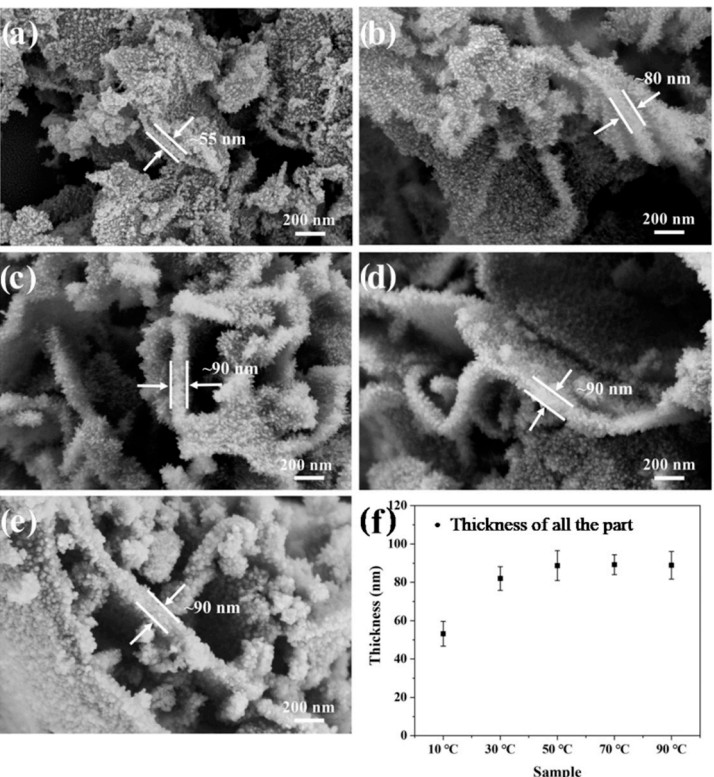

**Figure 7.** Typical SEM images of the samples with different synthesis temperatures: (**a**) 10 °C, (**b**) 30 °C, (**c**) 50 °C, (**d**) 70 °C and (**e**) 90 °C. (**f**) The thickness data of the samples.

Interestingly, the color of the prepared samples changed from white to light yellow as the synthesis temperature increased (Figure 8a). The UV–vis diffuse reflectance spectra of the samples, as shown in Figure 8b, demonstrate that the white samples show strong absorption in the UV region, and the optical absorption wavelength was consistent with that of $TiO_2$. The light yellow samples show strong absorption in the UV region as well as extra absorption in the visible light region (400–600 nm). The absorption peaks at 450 nm and 700–800 nm are attributed to the existence of the oxygen vacancy and the reduced $Ti^{3+}$ ions, respectively [35]. Obviously, simply increasing the synthesis temperature can enhance the absorption of the samples in the visible light region while keeping the phase composition unchanged as anatase $TiO_2$ (Figure S5). This strategy may find applications

in photocatalysis to address energy and environmental challenges through light-chemical conversion processes [36].

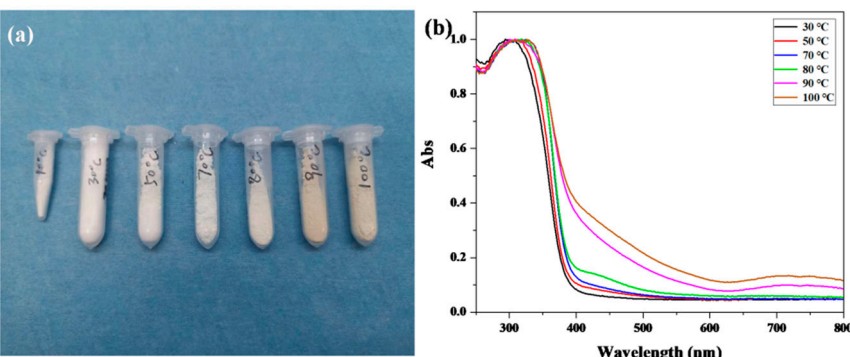

**Figure 8.** (**a**) Color change diagram of different samples, (**b**) typical UV–vis DRS of the sample with different synthesis temperatures.

## 4. Conclusions

In summary, we thoroughly studied the fabrication of $TiO_2$ nanoplates using $Ti_{0.87}O_2$ nanosheets via the NSG method. It is found that the concentration of the precursor and the synthesis time are the effective parameters in tuning the thickness of the obtained $TiO_2$ nanoplates. The synthesis time does not change the phase composition, while, as the concentration of the precursor increases, the phase of the prepared sample changes from anatase $TiO_2$ to $NH_4TiOF_3$, and the morphology of the sample will change from flake- to disc-shaped. In addition, it is found that the samples grow in the Stranski–Krastanov mode on the templates, and the growth follows second-order kinetics. Furthermore, the synthesis temperature has a great effect on the morphology and thickness of the sample but not on the phase composition of the sample. More importantly, the synthesis temperature can change the color of the sample and expand the light absorption range of the sample. However, the stirring rates have almost no effect on the morphology and composition of the sample.

**Supplementary Materials:** The following are available online at https://www.mdpi.com/article/10.3390/coatings12111673/s1, Figure S1: XRD of the samples with different synthesis time, 3 h, 6 h, 12 h, 24 h and 48h, Figure S2: AFM images of $TiO_2$ growth on monolayer $T_{0.87}O_2$ nanosheet thin film, (a) 2 h, (b) 4 h, (c) 8h and (d) 16h, Figure S3: Typical SEM images of the samples, (a) 0.025 M, (b) 0.05 M, (c) 0.1 M, (d) 0.15 M and (e) 0.2 M; (f) SEM image of the sample after calcination at 0.3 M, Figure S4: SEM images of the samples with different stirring rates, (a) 300 rpm, (b) 600 rpm, (c) 900 rpm, (d) 1200 rpm and (e) 1500 rpm; (f) XRD patterns of the samples, Figure S5: XRD of the samples with different synthesis temperatures, 10 °C, 30 °C, 50 °C, 70 °C, 80 °C, 90 °C and 100 °C.

**Author Contributions:** Conceptualization, H.Y.; Data curation, Y.Z., H.L. and J.C.; Funding acquisition, H.Y.; Investigation, Y.Z., H.L. and J.C.; Methodology, H.Y. and B.W.; Resources, X.B. and D.Y.; Supervision, X.B., D.Y. and H.Y.; Visualization, Y.Z. and H.L.; Writing—original draft, Y.Z. and H.L.; Writing—review & editing, H.Y. and B.W. All authors have read and agreed to the published version of the manuscript.

**Funding:** This research was funded by the Young Top-notch Talent Program of Zhengzhou University (No. 125/32310189), the China Postdoctoral Science Foundation (No. 2020M672267) and the National Natural Science Foundation of China (NSFC) (Grant. No. 51902290).

**Institutional Review Board Statement:** Not applicable.

**Informed Consent Statement:** Not applicable.

**Data Availability Statement:** Not applicable.

**Acknowledgments:** The authors would like to thank Tingting Xu for the TEM measurements, and they also appreciate Shawn Zhang's help with the language polishing.

**Conflicts of Interest:** The authors declare no conflict of interest.

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
