# Peer review of "Research on the Thickness and Microstructure of Plate-like TiO2 by the Nanosheet-Seeding Growth Technique"

_coatings, doi:10.3390/coatings12111673_

Round 1

Reviewer 1 Report

The work by Zhang et al dealing with  nanosheet-seeding growth TiO2 is interesting experimental work , where synthesis is presented.

Following should be improved: 

EDS analysis should be added to morphology study using SEM , 

TEM analysis for plates -like shapes crystallinity could be shown and would be additive for XRD.

the measurement of the thickness should be given as average , it is strange that e.g.Fig 5 thickness is always 80nm, how it was measured?

XRD is just showing phases evaluation , but changes in crystalline state of the samples could be  done , it is visible that for sample 0.3M is pattern more sharp, authors should re -thing some calculation of crystallite size, how the needles are presented measuring XRD?

The discussion towards applications should be done.

there are typos>(Fig. S4a-e)

Reviewer 3 Report

This is a quite interesting article that can be recommended for publication, but after clarifying and detailing some parts of the text.

1.     Introduction. It's clearly not enough. Only a few references refer to TiO2. The relevance of the work and a clear understanding of novelty requires additional justification.

More new information about TiO2 is required. In particular, how important is the structure and size of nanostructures? Furthermore, absolutely necessary to reflect what has been done in recent years. See, for example, few recent MDPI papers:

Pascariu, P.; Cojocaru, C.; Airinei, A.; Olaru, N.; Rosca, I.; Koudoumas, E.; Suchea, M.P. Innovative Ag–TiO2 Nanofibers with Excellent Photocatalytic and Antibacterial Actions. Catalysts 202111, 1234. https://doi.org/10.3390/catal11101234

Tsebriienko, T.; Popov, A.I. Effect of poly(titanium oxide) on the viscoelastic and thermophysical properties of interpenetrating polymer networks. Crystals 202111, 794.

https://doi.org/10.3390/cryst11070794

Badmus, K.O.; Wewers, F.; Al-Abri, M.; Shahbaaz, M.; Petrik, L.F. Synthesis of Oxygen Deficient TiO2 for Improved Photocatalytic Efficiency in Solar Radiation. Catalysts 2021, 11, 904. https://doi.org/10.3390/catal11080904

2.     It is not clear how justified the abbreviation TO ?

3.     Line 102. Diffraction is not sensitive to small concentrations of impurities, so this statement should be proved, for example, by the luminescent method, or the statement should be softened.

4.     Figure 1(h) and 2 require error bars, and corresponding discussion in the text.

5.     Figure 7(b). There is absolutely no discussion and corresponding analysis of the presented spectra.  Analysis of the absorption spectra at 450 and 700–800 nm is not presented.

Round 2

Reviewer 1 Report

Dear Authors, 

you have  read all your contribution to review and I can say that you have improved your work well.

Reviewer 2 Report

The Authors have revised manuscript according to suggestions. I recommend to accept the manuscript as is.

Reviewer 3 Report

The authors have significantly and successfully improved their original manuscript which how can be certainly recommended for publication.